# Olive Leaf Processing for Infusion Purposes

**DOI:** 10.3390/foods12030591

**Published:** 2023-01-30

**Authors:** Eva María Ramírez, Manuel Brenes, Concepción Romero, Eduardo Medina

**Affiliations:** Food Biotechnology Department, Instituto de la Grasa (IG), CSIC. Ctra. Utrera km 1, Building 46, 41013 Seville, Spain

**Keywords:** phenolic compounds, triterpenic acids, dehydration, oleuropein, olive tea, grinding

## Abstract

Olive leaf is a by-product rich in bioactive compounds, such as polyphenols and triterpenic acids, with numerous biological activities for human health. Nowadays, the existence of dry olive leaves marketed for infusion elaboration is lacking. During the elaboration process, the drying and grinding stages are critical for the conservation of bioactive compounds, and, precisely, the existing research on olive leaf production procedures is quite scarce. This work aimed to study and model the dehydration process using a forced-air oven and infrared with air convection systems. In addition, different grinding grades were studied. The kinetic constant and activation energy during dehydration were obtained. Drying temperatures above 50 °C produced a decrease in the phenolic concentration of olive leaves; however, it has been observed that prior storage of 24 h at room temperature considerably reduced the loss of phenols. Likewise, it was observed that the higher the degree of grinding, the greater the diffusion of both bioactive compounds and colored compounds. Therefore, the drying and grinding stages were closely related to the content of beneficial compounds and the appearance of the infusions, and their optimization was of crucial importance to produce dried olive leaves rich in biocompounds for use as healthy infusions.

## 1. Introduction

For many centuries, olive leaves have been widely used as a folk remedy in traditional medicine and their extracts have been associated with food preservation, cosmetics, and health. Further, they have been used in the human diet as an herbal extract tea or a powder supplement [1].

Olive leaves are considered a by-product of olive oil and table olive farming. However, olive leaves are also rich in bioactive compounds, such as polyphenols and triterpenic acids, that could contribute to the valorization of their by-product and to enhance the circular economy [2,3,4,5,6]. In the literature, there are many studies related to olive leaves, their extraction, analysis, health beneficial properties, and potential uses [7]. Thus, researchers have studied, in vitro and in vivo, numerous biological activities attributed to the bioactive compounds of the olive leaf and their beneficial properties for human health. Among them, the phenolic compounds have shown anti-hypertensive, antioxidant, hypoglycemic, hypocholesterolemic, and antimicrobial effects [6,7,8]. Likewise, triterpenic acids possess a wide spectrum of promising properties, such as antimicrobial, anti-tumor, anti-inflammatory, and anti-HIV, among other activities [6,8,9].

Oleuropein is the main phenolic compound in olive leaves and its concentration can reach between 6 and 9% of dry weight (DW) [1,3,10,11]. Other phenolic compounds identified in a lower concentration are hydroxytyrosol, tyrosol, caffeic acid, p-coumaric acid, verbascoside, vanillic acid, vanillin, luteolin, and rutin [3,12,13,14]. Moreover, oleanolic acid is the main triterpene in olive leaves, followed by maslinic acid [3,14], accounting for up to 20 g/kg of total triterpenic acids [15]. However, the concentration of phenolic and triterpenic compounds in leaves depends on several factors, including the origin, olive tree cultivars, climatic conditions, harvesting season, agricultural practices, etc. [2,4,14,16,17].

On the other hand, consumers are today demanding many different types of commercial herbal infusions (tea, peppermint, chamomile, etc.) as well as beverages based on these products. The production of dried olive leaves marketed for infusion purposes is insignificant, despite the positive perception that consumers have about olive products. Several reasons could explain this low exploitation of olive leaves as infusion herbs: (i) the scarce tradition of making olive leaf infusions, (ii) the presence of pesticides in non-organic olive leaves, (iii) the low quality of the commercial products, and (iv) lack of research to make attractive products from a sensory and nutritional point of view. Likewise, previous data obtained by Medina et al. [3] on the content of bioactive substances in commercial olive leaf infusions revealed great variability in these substances and quality.

The elaboration of olive leaf products for infusion purposes consists of a stage of washing, drying, grinding, and packaging. It is known that the leaf pretreatments before infusion substantially affect the amount of target compounds [7]. Undoubtedly, the drying treatment is the most critical stage in the process and it will greatly affect the final concentration of these bioactive compounds in the final product. During the drying process, cell tissues break down and the phenolic compounds come into contact with endogenous enzymes that can lead to the transformation of these substances [18,19].

Several treatments for olive leaf dehydration include solar, room temperature, microwave, and freeze drying [20], but studies on infrared drying are not available, which could reduce the drying time and labor for a continuous system. The drying temperature has a great influence on the content of biocompounds and high variability has been observed among the results obtained by previous authors [7,17], although none of them have modeled the drying stage. The final objective of many researchers was to obtain concentrated olive leaf extracts rich in biocompounds, mainly by using ultrasound-assisted, microwave, supercritical fluids, pressurized liquid, and others [6,21,22], but none considered obtaining dried leaves for the preparation of an olive tea; thereby, the optimization of the drying process has not been previously studied. In addition, researchers have focused their interest on the extraction of phenolic compounds, with very little information on the recovery of triterpenic acids from the leaf. In the case of dried olive leaves intended for leaf infusions, many variables could affect the content of the dried material production bioactive compounds, including the degree of grinding. Likewise, the diffusion of phenols and triterpenes to an aqueous phase has not been studied in depth. The standard times used in the preparation of infusions (5–10 min approx.) are a handicap compared to the long maceration times used by other researchers.

To our knowledge, no research is available on olive leaves for infusion elaboration. Therefore, this study aims to maximize the concentration of bioactive compounds in the dry olive leaf used for infusion purposes. For this objective, different drying treatments and grinding grades were studied to retrieve and transfer as much biocompound content as possible to the infusion, so that olive leaf tea with great benefits for consumer health will be achieved.

## 2. Materials and Methods

### 2.1. Raw Material

The olive leaves were harvested randomly from all parts of Manzanilla olive trees located in the olive tree garden at Instituto de la Grasa (Sevilla) during the 2021/2022 season, discarding those that presented damage. Phenolic compounds were analyzed immediately in a sample of fresh olive leaves to avoid oxidation. Subsequently, the rest of the leaves were dried directly or after a storage period of 24 h at room temperature.

Then, dried leaves were triturated to a powder using an ultra-centrifugal mill ZM200 (Retsch GmbH, Haan, Germany) and were kept at room temperature for further analysis.

The leaf moisture was determined by weighing 10 g of olive leaves and then oven drying at 105 °C to constant weight.

### 2.2. Drying Procedures

Dehydration of olive leaves (20 g) was carried out using two pieces of equipment: (i) an infrared with air convection IRCDi3HP-V of the IRconfort Company S. L. (Mairena del Aljarafe, Seville, Spain) and (ii) a forced-air laboratory oven (Dry-Big, Selecta, Spain) with an air velocity of 0.95 m/s. The infrared equipment had 3 chambers and 4 infrared heaters of 0.45 kW each. The distance between the heaters was 9 cm and the air velocity was 0.7 m/s and the perforated trays were placed at such a height that the leaves were equidistant from the heaters. The dehydration was performed at 40, 50, 60, and 70 °C in both systems, and the weight losses of samples were measured at different intervals.

The dehydration kinetics was adjusted to the modified Page model [23] with exponent n=2 as the following equation: $MR=a\exp\left(-kt^{2}\right)$, where $MR=M_{t}/M_{0}$. $M_{0}$ and $M_{t}$ are the initial moisture and moisture at time *t* (h), respectively. *k* is the drying coefficient in the Page model (s^−1^) and *a* is the dimensionless coefficient in the Page model.

The influence of the temperature on the drying kinetic constant of the Page model was evaluated using the Arrhenius equation: $\ln k=\ln A-E_{a}/RT$, where *k* is the Page constant, *R* is the gas constant, *T* is the drying temperature, and $E_{a}$ is the activation energy.

Once the dehydration process was modeled, fresh olive leaves were dried at 40, 50, 60, 70, and 80 °C for 15, 6, 3.5, 1.25, and 1 h, respectively, by the oven technology, and for 7.5, 3, 1.16, 0.5, and 0.5 h, respectively, using the infrared equipment. This experiment was carried out with olive leaves directly after harvesting or after a storage period of 24 h at room temperature.

### 2.3. Grinding Procedures

The dried leaf samples obtained from the different drying treatments were crushed in a ZM200 ultra centrifugal mill (Retsch GmbH, Haan, Germany) using a 6 mm-diameter crushing sieve. The shredded samples obtained were used for subsequent chemical analysis and for the infusion preparation, as described below.

Likewise, an experiment was carried out to study the degree of grinding. In this case, the leaves of the Manzanilla cultivar (200 g) were dried using infrared equipment at a temperature of 80 °C for 40 min. Subsequently, the leaves were crushed with the ultra centrifugal mill using the 10 mm-diameter crushing sieve. This shredded sample was passed through various sieves with different pore diameters (5, 3.15, 2, 1, and 0.5 mm) to obtain different grinding fractions. Then, the infusions were prepared as described below, and the phenolic compounds, triterpenes, color, and turbidity of the infusions were analyzed.

### 2.4. Olive Tea Preparation

The olive tea elaboration consists of directly mixing 1.7 g of dried and ground leaves with 240 mL of boiling water for 5 min with occasional shaking. The content of phenolic compounds in the infusions was analyzed immediately without any storage period.

### 2.5. Phenolic Compound Analysis

The phenolic compounds in the olive leaves were analyzed as described elsewhere [3]. Two grams of olive leaves (fresh or dried) was mixed with 30 mL of dimethyl sulfoxide (DMSO), homogenized in Ultra-Turrax homogenizer (Ika, Breisgau, Germany), and sonicated in an ultrasound bath for 10 min. After 10 min of resting contact, the mixture was centrifuged at 9000× *g* for 5 min. The supernatant (0.25 mL) was diluted with 0.5 mL of DMSO and 0.25 mL of internal standard (0.2 mM syringic acid in DMSO). Likewise, an aliquot of 0.25 mL of olive leaf infusion (previously acidified with phosphoric acid) was mixed with 0.5 mL of water and 0.25 mL of internal standard (0.2 mM syringic acid in water). Then, samples were filtered through a 0.22 μm-pore-size nylon filter, and an aliquot (20 μL) was analyzed via high-performance liquid chromatography (HPLC).

The chromatographic system consisted of a Waters 717 plus autosampler, a Waters 600 E pump, a Waters heater module, and a Waters 996 photodiode array detector operated with Empower 2.0 software (Waters Inc., Milford, MA, USA). A Spherisob ODS-2 column (Waters Inc.) at 35 °C with a flow rate of 1 mL/min was used for the analysis. The separation was achieved by gradient elution using water (pH 2.5 adjusted with phosphoric acid) and methanol with an initial composition of 90% and 10%, respectively. Phenolic compounds were monitored at 280 nm. Each compound was evaluated using a regression curve with the corresponding standard. All analyses were performed in triplicate.

### 2.6. Triterpenic Acid Analysis

Extraction of triterpenic acids from olive leaves was performed as described by Romero et al. [14] with slight modifications. Half a gram of dried olive leaf powder was mixed with 4 mL of methanol/ethanol (1:1, *v*/*v*), vortexed for 1 min, centrifuged at 9000× *g* for 5 min at 20 °C, and the solvent was separated from the solid phase. This step was repeated six times, and the pooled solvent extract was vacuum-evaporated. Subsequently, the residue was dissolved in 4 mL of methanol and filtered through a 0.2 µm-pore size.

The analysis of triterpenes in olive leaf tea was performed by mixing 10 mL of the infusion with 10 mL of ethyl acetate. The mix was vortexed for 1 min, centrifuged at 9000× *g* for 5 min, and the solvent phase was separated from the aqueous phase. This step was repeated six times, and the pooled solvent extract was vacuum-evaporated. Subsequently, the residue was dissolved in 1.33 mL of methanol and filtered through a 0.2 µm-pore size.

An aliquot of 20 µL of the sample was used for HPLC analysis. The chromatographic system and column were the same as those used for the phenolic compound analysis. Elution was performed at 35 °C with a mobile phase of methanol: acidified water with phosphoric acid at pH 3.0 (92:8, *v*/*v*), at a flow rate of 0.8 mL/min, and the eluate was monitored at 210 nm. Oleanolic and maslinic acids were quantified using external standards (Sigma, St. Louis, MO, USA). All analyses were performed in triplicate.

### 2.7. Color and Turbidity Analysis

The color of olive leaf tea was measured using a Shimadzu UV-vis 1800 spectrophotometer (Kyoto, Japan), equipped with computer software to calculate the CIE L* (lightness), a* (redness), b* (yellowness), hue angle, and chromaticity parameters. The turbidity of olive leaf tea was measured using a Hach turbidimeter 2100A (Iowa, USA). All analyses were performed in triplicate.

### 2.8. Statistical Analysis

Statistica software 7.0 (StatSoft, Inc., Tulsa, OK, USA) was used for data analysis. Data were expressed as mean values ± standard deviation. Statistical comparisons were performed by one-way analysis of variance (ANOVA) followed by Duncan’s multiple range test. A value of *p* < 0.05 was considered statistically significant.

## 3. Results and Discussion

The changes in the moisture and drying rate of olive leaves at temperatures of 40, 50, 60, and 70 °C are given in Figure 1. The drying kinetic of olive leaves was influenced by the temperature and the drying equipment used. The average initial moisture content was 49.6 ± 1.9%. As expected, dehydration time decreased at higher temperatures, irrespective of the heating equipment. Total moisture loss was reached at 15.0, 6.0, 3.7, and 1.3 h at air temperatures of 40, 50, 60, and 70 °C, respectively, for the drying at the forced-air oven (Figure 1A). The drying in the infrared equipment managed to reduce the dehydration time considerably, being only necessary 7.5, 3.0, 1.17, and 0.58 h at temperatures of 40, 50, 60, and 70 °C (Figure 1B). Boudhrioua et al. [24] concluded that infrared drying induces considerable moisture removal from fresh olive leaves and short drying durations.

The experimental data obtained in this work fitted the modified Page model [23] with coefficients of determination (*R*^2^) higher than 0.98. The drying rate (*k*) increased at the same time as the temperature rose, obtaining higher coefficients for dehydration (*a*) in the infrared equipment than in the oven (Appendix A), so the time required for drying the leaves was much shorter. Temperature dependence of the drying kinetic constant has been shown to follow an Arrhenius relationship, with *R*^2^ values of 0.99 and 0.97, and the activation energies (*E_a_*) were 139.66 and 166.62 kJ/mol for oven and infrared dehydration, respectively (Figure 2).

Figure 3 shows the phenolic composition in the olive leaves before and after the different heat treatments carried out. The total phenolic concentration in the fresh leaf before dehydration was 13,276 mg/kg DW when it was analyzed immediately after harvesting and 14,575 mg/kg DW when left for 24 h at room temperature. As expected, the main polyphenol in the olive leaf was the bitter compound oleuropein, which constitutes more than 90% of the total phenols [17]. This substance, together with the rest of the phenolic compounds, decreased the concentration after receiving the dehydration treatments. The greatest losses were observed when leaves were dried immediately after harvesting at temperatures of 60, 70, and 80 °C in the oven, and 50 and 60 °C in the infrared, reducing the total phenols to a concentration between 300 and 4000 mg/kg DW, which represents a loss between 97 and 70% of the initial concentration. Similar results were obtained by Feng et al. [25], who found a higher concentration of oleuropein at 45 °C than at 70 °C during oven drying of olive leaves. Boudhrioua et al. [24] also reported that total phenol content decreased at high temperatures (T ≥ 60 °C) in Chetoui and Chemchali blanched leaves, and Helvaci et al. [26] defined the optimal drying conditions at 50 °C for the minimum losses of determined quality parameters, recovering 88.8% of the total phenolic content. Likewise, Sahin and Bilgin [7] reported that drying temperatures from 65 to 80 °C resulted in a declining concentration of oleuropein in the leaves. Enzymatic activities might be responsible for this decrease. During the drying process, cell tissues break down and the phenolic compounds come into contact with endogenous enzymes that can lead to the degradation of these substances [18,27].

On the other hand, the losses were lower when a temperature of 40 °C was used, assuming a decrease in total phenols of 32%, not finding significant differences for both heating systems used. However, the lowest loss in phenolic concentration (14–65%) occurred when the leaves were kept at room temperature for 24 h before the drying treatment (Figure 3), obtaining a higher phenolic content than those dried immediately after harvesting.

The dehydration process is crucial in the elaboration of leaves for infusions. Among all the methods, open-air sun drying is not an appropriate way of drying herbal plants due to several drawbacks, such as contamination with dust and insects and uncontrollable drying parameters, which can cause over-drying as well as loss of quality in the dried product. Therefore, drying in a closed and controlled environment should be preferred [26]. Traditionally, air-drying is the most employed procedure, being easy and economic. However, air-drying is more time-consuming compared to oven- and infrared-drying. The effect of oven-drying is unclear; some studies have presented polyphenol degradation while other investigations concluded that drying olive leaves at high temperatures for a short time did not degrade their polyphenols [28]. In this study, the most effective treatment was the one carried out with leaves that remained for 24 h at room temperature and the drying was carried out in an oven at a temperature of 40 °C (Figure 3). With the previous conditions, the loss of phenols was below 15%. However, it must be taken into account that, for these conditions described, the drying time was 15 h (Figure 1), and this time can be reduced by half when infrared was used, assuming energy savings, although increasing the loss of phenols up to 35%.

The drying treatments of the olive leaves also influenced the color of the infusions made from them. The color parameters are shown in Table 1. The infusions made with leaves dried at 40 °C, regardless of the system used, showed a higher value of the luminosity L*, which indicated a significantly lighter color than the rest of the infusions, which showed a darker color. Likewise, the low drying temperatures showed the lowest values of parameter a*, indicating a greener tone in the infusion. As the drying temperature in the leaves increased, the values of a* were higher, denoting a more reddish-brown tone. Again, lower values of parameter b* were found for drying carried out at 40 °C, showing more yellow tones as the temperature increased. These results confirmed those obtained by Boudhrioua et al. [24], finding that the increase in the dry temperature by infrared increased the values of the parameters a* and b* in dried olive leaves. In this study, Manzanilla leaves showed a clear trend for the chromaticity parameter, showing higher values as the drying temperature increased, being the highest for the infusions prepared with the leaves dried at 80 °C in the oven. No trend was found for the Hue angle parameter.

In addition to the influence on the degradation of phenols during the drying stage, the grinding step is another important factor in olive leaf processing. Table 2 shows the influence of grinding degree on the content of oleuropein, triterpenic acids, and the color of the infusions prepared with these olive leaves. The diffusion of phenols, and in particular of oleuropein as the main phenol, to the infusion water was greater as the degree of grinding was lower. Islam et al. [29] also observed a higher polyphenol concentration in the infusions when medicinal herbs (cardamom, cinnamon, clove, nigella, and ginger) were ground compared to whole herbs. Alsaud and Farid [30] reported that the extraction of biocompounds of Manuka leaves was faster when a fine powder was employed and indicated that the release of bioactive compounds occurred during grinding. Zhang et al. found that smaller particle sizes had better infusion properties for *Lycium ruthenicum* [31]. Shu et al. [32] also observed a greater amount of bioactive compounds in green tea powder as particle size decreased with increased bioaccessibility.

In this study, the highest diffusion of oleuropein was obtained from the leaves with a grinding grade < 0.5 mm, reaching concentrations of approximately 900 mg/kg, which represents around 10% of the initial content of the dried olive leaf. In contrast, oleanolic and maslinic acids did not diffuse at grinding degrees above 1 mm, and low diffusion to the infusion water, less than 0.5%, for the rest. This fact was previously observed by Medina et al. [3] and it is due to the low solubility of these compounds in aqueous phases at neutral pH.

The degree of grinding also influenced the color parameters of the infusions (Table 2). The values of parameters a* and b* increased as the degree of grinding was lower, the infusions being more reddish and yellowish, respectively. Likewise, the chromaticity and the turbidity also increased when grinding degree decreased, finding significant differences among them. On the contrary, the luminosity L* and the Hue angle decreased as the degree of grinding was lower, with the infusions being darker. Therefore, it was evident that the diffusion of both the bioactive compounds and the colored pigments in the infusion water was higher with decreasing degrees of grinding. This fact was also reported when olive leaves were added to the production of olive oil. The colored compounds of the leaves, main chlorophyll, and carotenoid compounds increased the green color of the olive oils obtained [33].

## 4. Conclusions

The design of the dehydration process of olive leaves was modeled using a forced-air oven and infrared equipment, and the kinetic constant and activation energy values were calculated. In this study, the most effective treatment was the one carried out with leaves that remained for 24 h at room temperature and dried in an oven at a temperature of 40 °C for 15 h, recovering 85% of the initial polyphenols. However, the dehydration time can be reduced by half when infrared is used, assuming energy savings and a continuous system, although increasing the loss of phenols.

Moreover, the degree of grinding of the olive leaves greatly influenced both the diffusion of bioactive compounds and the color of the infusion. The higher the degree of grinding, the greater the diffusion of compounds, giving rise to a colored olive infusion, rich in beneficial compounds for the health of the consumer. Additionally, color parameters indicated a clear relationship with the different drying treatments, and a panel test will be studied in the future to determine consumer preferences.

## Figures and Tables

**Figure 1 foods-12-00591-f001:**
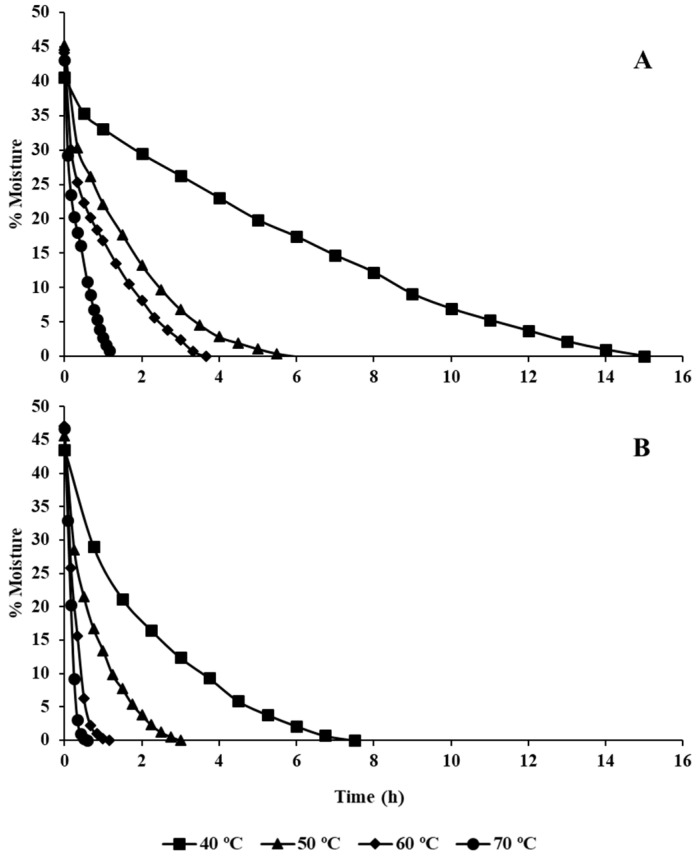
Relationship between the percentage of moisture (%) and drying time (h) at different temperatures. Panel (**A**): drying in the oven. Panel (**B**): drying in the infrared equipment.

**Figure 2 foods-12-00591-f002:**
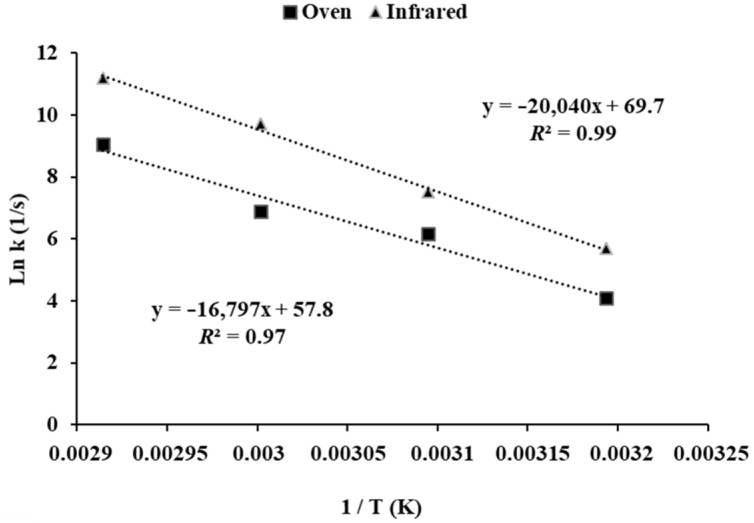
Arrhenius-type relationships between the drying constant rate and temperature obtained during the dehydration of olive leaves using the oven or infrared heating.

**Figure 3 foods-12-00591-f003:**
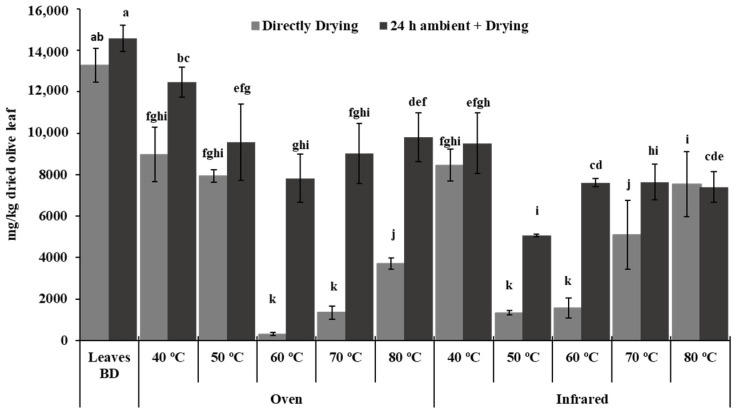
Polyphenol concentration (mg/kg dried olive leaf) after different heat treatments. Leaves were directly dried after harvesting or after a storage period of 24 h at ambient temperature. All the heat treatments were performed until the weight was constant. Data are expressed as the mean value of triplicate. Bars mean the standard deviation. BD means before dehydration. Vertical bars with different lowercase letters indicate significant differences according to Duncan’s multiple range test (*p* < 0.05).

**Table 1 foods-12-00591-t001:** Influence of drying temperature on the color of the olive leaf infusions. Data are expressed as the mean value of triplicate. Standard deviation in parentheses. Different lowercase letters indicate significant differences according to Duncan’s multiple range test (*p* < 0.05).

Equipment	Temperature (°C)	L*	a*	b*	Chromaticity	Hue Angle
Oven	40	98.3 (0.4) ^b^	1.4 (0.0) ^f^	9.3 (0.1) ^e^	9.5 (0.1) ^ef^	81.7 (0.1) ^c^
	50	97.4 (0.2) ^d^	2.5 (0.0) ^ab^	15.5 (0.2) ^ab^	15.7 (0.2) ^ab^	80.8 (0.1) ^fg^
	60	97.8 (0.3) ^cd^	2.3 (0.1) ^bc^	14.4 (0.7) ^bc^	14.6 (0.7) ^bc^	81.0 (0.1) ^ef^
	70	97.6 (0.8) ^cd^	2.1 (0.3) ^cd^	13.3 (1.3) ^cd^	13.5 (1.4) ^cd^	81.2 (0.4) ^ef^
	80	98.0 (0.2) ^bc^	1.7 (0.0) ^e^	12.8 (0.0) ^d^	12.9 (0.0) ^d^	82.5 (0.0) ^b^
Infrared	40	99.0 (0.1) ^a^	1.0 (0.1) ^g^	7.7 (0.3) ^f^	7.8 (0.3) ^fg^	82.8 (0.1) ^a^
	50	98.1 (0.1) ^bc^	1.9 (0.1) ^de^	13.5 (0.5) ^cd^	13.6 (0.5) ^cd^	81.9 (0.0) ^c^
	60	97.6 (0.2) ^cd^	2.4 (0.2) ^b^	14.4 (0.1) ^bc^	14.6 (1.0) ^bc^	80.7 (0.0) ^g^
	70	97.3 (0.1) ^d^	2.7 (0.2) ^a^	16.7 (0.9) ^a^	16.9 (1.0) ^a^	80.8 (0.0) ^fg^
	80	97.3 (0.3) ^d^	2.5 (0.3) ^ab^	16.7 (2.0) ^b^	16.9 (2.1) ^a^	81.5 (0.1) ^d^

**Table 2 foods-12-00591-t002:** Influence of the grinding grade on the oleuropein and triterpene content (mg/kg DW), color, and turbidity of the olive leaf infusions. Data are expressed as the mean value of triplicate. Standard deviation in parentheses. Different lowercase letters indicate significant differences according to Duncan’s multiple range test (*p* < 0.05). ND: not detected.

Sample	Grinding Grade (mm)	Oleuropein (mg/kg)	Oleanolic Acid (mg/kg)	Maslinic Acid (mg/kg)	Color Parameters					Turbidity (NTA)
					L*	a*	b*	Chromaticity	Hue Angle	
Dried leaves	-	9419 (955) ^a^	28,264 (2547) ^a^	5755 (206) ^a^	-	-	-	-	-	-
Leaves infusions	>5	42 (12) ^f^	ND	ND	98.7 (0.4) ^a^	−0.8 (0.0) ^e^	−1.3 (0.1) ^f^	1.5 (0.1) ^d^	237.8 (1.0) ^a^	2.0 ^f^
	5–3.15	80 (2) ^f^	ND	ND	98.6 (0.3) ^a^	−0.7 (0.0) ^e^	−0.7 (0.2) ^e^	1.0 (0.2) ^e^	221.2 (8.9) ^b^	2.5 ^e^
	3.15–2	159 (3) ^e^	ND	ND	97.9 (0.3) ^ab^	−0.6 (0.0) ^d^	0.7 (0.2) ^d^	0.9 (0.1) ^e^	129.4 (9.1) ^c^	6.5 ^d^
	2–1	358 (0) ^d^	ND	ND	97.6 (0.9) ^bc^	−0.2 (0.1) ^c^	3.5 (0.2) ^c^	3.5 (0.2) ^c^	93.9 (1.6) ^d^	9.0 ^c^
	1–0.5	727 (24) ^c^	1 (0) ^b^	1 (0) ^b^	97.1 (0.7) ^c^	0.3 (0.1) ^b^	9.0 (0.7) ^b^	9.0 (0.7) ^b^	88.4 (0.3) ^d^	15.0 ^b^
	<0.5	906 (30) ^b^	14 (0) ^b^	4 (0) ^b^	95.8 (0.3) ^d^	0.8 (0.0) ^a^	14.5 (0.2) ^a^	14.5 (0.2) ^a^	86.8 (0.0) ^d^	35.0 ^a^

## Data Availability

The data are contained within the article.

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
