# Peer review of "Olive Leaf Processing for Infusion Purposes"

_foods, 2023, doi:10.3390/foods12030591_

Round 1

Reviewer 1 Report

Table 2 did not fit you!

Manuscript Olive leaf processing for infusion purposes addressed on by-product olive leaf as a rich source of bioactive compounds. As there is a lack of dry leaves on the market, in this study they deal with the comparison of different dehydration processes, of forced air ovens and infrared air convection systems, different grinding grades, all with the aim of obtaining the best quality of preserved leaves. The topic of this work is not new, but it provides new useful information about ways of drying olive leaves to improve product quality. Olive leaves were dried at different temperatures: 40, 50, 116 60, 70 and 80 ºC and in  different period of time : 15, 6, 3.5, 1.25 and 1 hour, respectively by the oven technology, and 117 for 7.5, 3, 1.16, 0.5 and 0.5 hours, respectively using the infrared equipment. They obtain that drying temperatures above 50 ºC produced a decrease in the phenolic concentration of olive leaves; however, it has been observed that prior storage of 24 hours at 15 room temperature considerably reduced the loss of phenols. They also observed that the greater the degree of grinding, the greater the diffusion were of both bioactive compounds and colored compounds. All above mentioned represents original contribution of this study. Conclusions followed results obtained in this manuscript, references correspond to statements in the text. After all my opinion is that this study is interesting for the production of olives that can use this part of the olive, especially in periods of significant pruning of these trees.

Reviewer 2 Report

This work aimed to study and model the dehydration process by a forced air oven and an infrared with air convection systems. In addition, different grinding grades was studied. The kinetic constant and activation energy during the dehydration were obtained. Drying temperatures above 50 ºC produced a decrease in the phenolic concentration of olive leaves, however, it has been observed that prior storage of 24 hours at room temperature considerably reduced the loss of phenols. Likewise, it was observed that the higher degree of grinding, the greater diffusion of both bioactive compounds and colored compounds. Therefore, the drying and grinding stages were closely related to the content of beneficial compounds and the appearance of the infusions, and their optimization was of crucial importance to produce dried olive leaves rich in biocompounds for use as healthy infusions.

  In this paper, only two processing methods or models were selected, and polyphenols and triterpenoids were analyzed. The results can not provide comprehensive scientific data for the processing of olive leaves.

Reviewer 3 Report

The work is very good but needs to be corrected. 

The data in table 2 is truncated. I suggest splitting it into two. 

The symbol for degrees Celsius is underlined 

The equations should be written in the editor 

"The average initial moisture content was 49.6 ± 1.9%" -this should be in the results 

The lack point is that the authors do not explain why the degree of grinding affects the color. The discussion should be rebuilt and conducted more deep.  

Alsaud, N.; Farid, M. Insight into the Influence of Grinding on the Extraction Efficiency of Selected Bioactive Compounds from Various Plant Leaves. Appl. Sci. 202010, 6362. https://doi.org/10.3390/app10186362

Shu, Y., Li, J., Yang, X., Dong, X., & Wang, X. (2019). Effect of particle size on the bioaccessibility of polyphenols and polysaccharides in green tea powder and its antioxidant activity after simulated human digestion. Journal of food science and technology56(3), 1127-1133.

Zhang, J., Dong, Y., Nisar, T., Fang, Z., Wang, Z. C., & Guo, Y. (2020). Effect of superfine-grinding on the physicochemical and antioxidant properties of Lycium ruthenicum Murray powders. Powder technology372, 68-75.

Round 2

Reviewer 2 Report

You have fully explained my doubts about the paper. I agreed to accept the paper.